# Enhancement of state response capability and famine mitigation: A comparative analysis of two drought events in northern China during the Ming Dynasty

Fangyu Tian[12], Yun Su[12], Xudong Chen[12], Le Tao[12]

[1] Faculty of Geographical Science, Beijing Normal University, Beijing, 100875, China
[2] Key Laboratory of Environmental Change and Natural Disaster, Ministry of Education, Beijing Normal University, Beijing, 100875, China

*Correspondence to*: Yun Su (suyun@bnu.edu.cn)

**Abstract.** Studying social impacts and responses to historical extreme climate events can offer valuable insights into coping with major disaster events and adapting to climate change better. This paper developed a model of the processes and responses to extreme drought-induced famines in ancient China. Based on this, the study explored the differences in famine causation and response effectiveness between Chenghua Drought (1483-1486 CE) and Wanli Drought (1585-1588 CE). The findings are as follows: (1) By the time of Wanli Drought, the increase in land reclamation had enhanced societal defence, preventing many drought-affected counties from experiencing famine. Even in cases where famines did occur, their severity was lower than during the Chenghua Drought. (2) State emergency measures, including exemption and relief, proved effective in mitigating famines. The stronger finance and economy during the Wanli Drought enabled more robust relief efforts, resulting in lower famine severity. (3) Famine response capabilities varied regionally. Shandong, Beijing, Tianjin and Hebei demonstrated strong defensive capabilities and effective state emergency responses, while Shanxi had weaker defensive abilities, making it more vulnerable to famine. The defensive capabilities in Henan, Shaanxi, Gansu, and Ningxia showed significant improvement.

## 1 Introduction

In historical periods with low agricultural productivity, extreme drought events had wide and profound impacts on human societies. Scholars have qualitatively described the link between human social development and climate change by identifying historical cases based on paleoclimate reconstruction. They argue that the disappearance of civilizations (Peterson and Haug, 2005), social unrest (Atwell, 2002; Zhang et al., 2005), and dynasty succession (Atwell, 2001; Zheng et al., 2014b) in various regions and countries in human history are temporally synchronized with extreme drought events. For instance, the rapid collapse of the Akkadian civilization in the Mesopotamian region, the Moche culture in northern Peru, and the Tiwanaku culture in South America coincides with periods of severe climate aridity (deMenocal, 2001) as revealed by reconstructions from natural proxies such as lake sediments and ice cores. Additionally, quantitative studies using

methods such as correlation and multiple regression analysis have demonstrated the connections between extreme drought events and production, economic and demographic systems including food availability (Hao et al., 2020), food prices (Brázdil et al., 2019), population migration (Pei and Zhang, 2014), and social unrest (Xiao et al., 2013; Yu et al., 2004), providing scientific arguments and empirical evidence for understanding the relationship between historical extreme drought events and human societal development (Pei, 2017).

The transmission process of the social impacts and responses of extreme drought events exhibits hierarchical and cascading effects (Zhang et al., 2011; Zheng et al., 2014b), implying that effective response actions have the potential to interrupt the transmission chain and mitigate the impacts of extreme drought. Conceptual models and causal chains reveal that when the impact of extreme drought events exceeds the capacity or resilience of a certain level, it is transmitted to the up level (Chen et al., 2021; Tao et al., 2024). The primary transmission pathway involves changes in resources and disasters that affect

agricultural production, subsequently influencing economic system such as food supply and population-related factors including migration, famine, and social unrest (Fang et al., 2014). Generally, the more severe the extreme drought event, the higher the level it affects and the longer the chain of consequences. Extreme drought events and the impacts, along with multi-level responses from human societies, exhibit a coupled process at various spatial-temporal scales, forming a network of interactions between extreme climatic events and society. Recent studies have attempted to identify key events or critical

nodes in this network that are influenced by climate change and have profound impacts on society through the study of complex networks (Chen et al., 2022). Targeted interventions that can alter these key nodes are likely to interrupt the transmission of the impacts of extreme climate events. At the same time, it is also implied that under different socio-economic conditions, the implementation of different responses or intervention measures can result in varying consequences for extreme climate events of the same magnitude (Tian et al., 2022, 2024). Human systems are socio-ecological or human-

environment coupled systems, with interactions between social subsystems and ecological subsystems. Typically, the transmission process of impacts is influenced by human society, which adopts responses based on socio-economic conditions, resulting in feedback processes in various domains such as production, economy, population, and society. These can either mitigate or amplify the consequences of extreme events (Engler et al., 2013; Su et al., 2018), until a new equilibrium is achieved between human society and the natural environment.

Famine is one of the consequences of extreme drought events and a critical link in the impact transmission process of extreme drought (Chen et al., 2024). Reducing the impacts of famines, represents a common global challenge throughout history and into the present. As it stands, global food security remains on high alert, with 680 million people grappling with inadequate food consumption and 16 countries unable to break free from very high levels of hunger (HungerMap LIVE, 2024). Additionally, food security in large cities is also a significant concern. These cities, with their large populations and

substantial resource demands, exhibit a distinct outsourcing characteristic in their food systems, that is consuming internally while relying heavily on external supplies. Extreme weather events can disrupt resource transportation and trigger panic buying, posing famine risks even in large urban (Zhang et al., 2023). Moreover, according to the IPCC Sixth Assessment Report, the majority of the world's inhabited areas are projected to experience increasingly frequent and severe extreme

events with global warming. Moreover, agricultural and ecological droughts are expected to intensify and become more complex (Calvin et al., 2023). Consequently, extreme droughts will continue to pose the obstacle to Zero Hunger for a long time to achieve (Shi et al., 2022).

In the long-standing battle against famine, human societies have accumulated a wealth of experience, effective adaptation and response measures. Given that famine often stems from poor harvests, it can be prevented by increasing cultivated land area, optimizing crop planting structures, and constructing irrigation systems to maintain or enhance agricultural productivity, thereby increasing food production and ensuring food security (Deng, 2011). Additionally, policies such as reducing agricultural taxes and prohibiting food exports can ensure regional food self-sufficiency (Nyamwange, 1995; Shiue, 2005) and reduce the exposure and risk of famine in affected areas. When famine does occur, policies, economic relief, and humanitarian assistance can be used to transfer grain from surplus regions to deficit areas, balance food market, or manage population migration-effectively mitigating the impact of the famine and preventing more serious social crises (Mishra and Singh, 2010). Hence, case studies on extreme drought events and their responses, alongside understanding the famine mechanisms, can provide valuable insights and replicable experiences for countries or regions at risk of food shortage.

During the Ming Dynasty (1368-1644 CE), natural disasters were frequent, especially in the mid- to late-Ming period (1435-1644 CE) (Bai and Wang, 2004, p.214). This phase, which coincided with the Little Ice Age (Ren et al., 2024), saw an increase in severe droughts across the eastern monsoon China (Hao et al., 2020), severely impacting agriculture, population and economy. By this time, the Ming Dynasty had reached a mature stage of feudal society, and emperors consistently prioritized famine mitigation as a core national policy (Ye, 1996). Famine mitigation policies during the Ming Dynasty were highly developed. And they also served as a key indicator of the state governance capacity (Chen, 2015). Research on the impacts and responses to these extreme drought events during the Ming Dynasty can enhance our understanding of disaster response mechanisms and offer valuable insights from historical experience of China for global disaster reduction.

Chenghua Drought (1483-1486 CE) and Wanli Drought (1585-1588 CE), which mainly affected the northern China, resulted in severe famines (Zhang, 2004). During the Chenghua and Wanli Droughts, there were few country-scale wars to consume national food resources or divert the attention of governmental administrations. Consequently, the central government implemented numerous famine relief measures and responded actively to both the two drought events. The social and economic contexts of the two drought events differed, however. Wanli Drought occurred after the state financial reforms, which mitigated the fiscal crisis and substantially improved the national economy, which could provide more food and money for responses to famine (Tian et al., 2024). Additionally, famine relief system was also continuously improved from the Chenghua to Wanli. Therefore, the comparative analysis of the droughts, famines and social responses during the two drought events can enhance our understanding of how institutional and economic factors play roles in mitigating famines. Such insights are vital for contemporary countries in strengthening the capacity to respond to extreme climate events and ensuring the food security of citizens.

## 2 Data sources and research methods

### 2.1 Overview of the study region

The two drought events primarily occurred in northern China during the Ming Dynasty, with the study area bounded by 31°-42°N and 100°-122°E. The study region encompasses the Loess Plateau and North China Plain, which were the earliest regions of agricultural development in ancient China. In the Ming Dynasty, the main crops cultivated in this region were wheat and millet, followed by rice and beans (Li, 2014; Lv, 2000). It fell under the category of dryland farming areas. Influenced by monsoon, precipitation in the study region is unevenly distributed across seasons, with precipitation from May to September accounting for more than 80% of the annual total. Precipitation shows significant interannual variability, leading to frequent drought and flood disasters and unstable agricultural yields. Furthermore, the population in northern China experienced a rapid increase in the 16th century and by 1630, the total population in the north had increased to about 62 million (Cao, 2000, p.451–452). The provinces in the North China Plain, such as Beizhili, Shandong, Shanxi, and Henan, were densely populated regions at that time, experiencing substantial population pressure (Liu, 2005; Zhang, 2006). Beizhili, being the political centre, and the northern parts of Shanxi and Shaanxi serving as military strongholds, required a supply of millions Dan[1] of grain annually, which magnified the contradiction between grain supply and demand.

During the process of information extraction and mapping in this study, historical place names within the study area were converted to present-day names, and modern maps were used. A total of 308 counties were involved in the study, and the study region is shown in Figure 1.

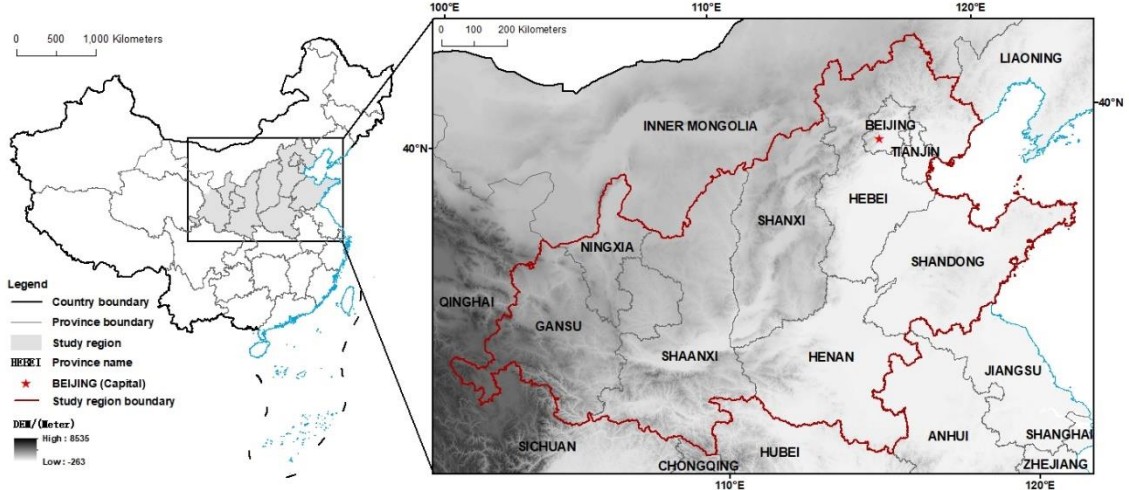

**Figure 1: A map of the study region. The digital elevation data were derived from Shuttle Radar Topography Mission (SRTM) datasets.**

---

[1] Dan (石) is a unit of volume in ancient China, and in the Ming Dynasty, the weight of 1 Dan grain is about 60~70 kg.

## 2.2 Research model and data sources

### 2.2.1 Drought-induced famine and its responses in ancient China

The processes of drought-induced famine and its responses in ancient China serves as the cornerstone of this paper. This field has been extensively explored and deeply understood by many scholars (Figure 2).

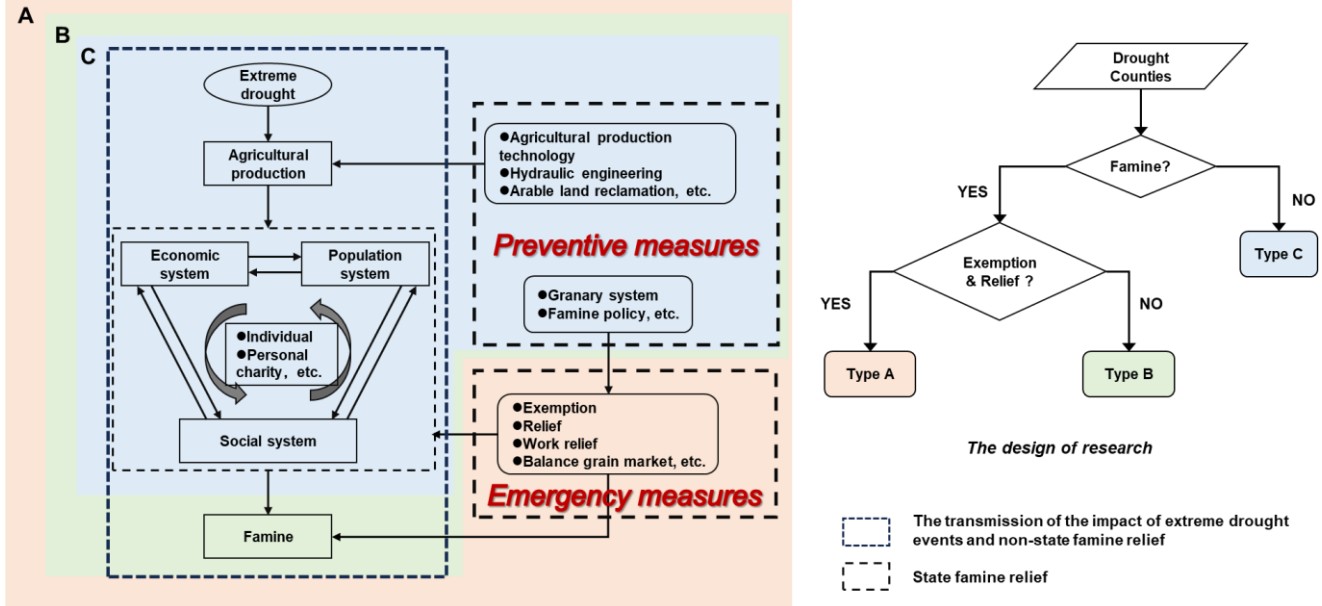

**Figure 2: The model of processes and responses to extreme drought-induced famines in ancient China and research design**

The process by which extreme droughts lead to famines is hierarchical. Numerous historical cases demonstrate that poor harvests and the collapse of agricultural production are the initial and most critical stages in this process (Baek et al., 2020; Engler et al., 2013). In ancient China, the primary source of food was agriculture (Bu, 2007). When extreme drought events persisted for many years, the basic means of obtaining food from agricultural production almost collapsed, leading to a sharp decline in individual food security. This breakdown in agricultural production would subsequently trigger a rise in food prices, a deterioration in public health, and a surge in social unrest, including theft and robbery, resulting in widespread disorder across economic, demographic, and social systems (Brázdil et al., 2018, 2019; Hao et al., 2021; Xiao et al., 2015; Zheng et al., 2014a). These posed stress to the state and society in maintaining food distribution and consumption security. When food production, distribution, and consumption could no longer be secured, famine became inevitable (Figure 2).

In response to prolonged and widespread droughts, individuals and families would initially attempt to cope with food shortages by relying on household food reserves and local markets. However, given the underdeveloped productivity in ancient China, food production was low, and the commercialization of food was limited, leaving poor farmers with little

capacity to cope with natural disasters (Guo, 2009; Ye, 1998). Occasionally, local gentry would donate money and grain to mitigate famine. But before the late Qing Dynasty, these private efforts generally served as a supplement to state-led relief (Li, 1993; Zhao, 2007). Overall, famine responses during the Ming Dynasty were predominantly government-led, with state relief being crucial for maintaining regional food security, ensuring individual food security for disaster victims, and mitigating famines. Consequently, this study focuses primarily on state relief.

Ancient China gradually developed a comprehensive famine response system (Figure 2). From the state perspective, these responses could be categorized into preventive and emergency measures, commonly referred to by historians as famine preparedness and famine relief (Deng, 2011, p.166–217). Preventive measures were implemented before the onset of famine, focusing on increasing food production and reserves (Shiue, 2005). The central government encouraged peasants to expand arable land, introduced high-yield and drought-resistant crops, constructed hydraulic engineering, and supported other methods to enhance land use efficiency, thereby securing food production (Deng, 2011, p.202–217). Additionally, the government paid more attention to building granaries and improving famine relief policies to ensure sufficient food supplies and effective relief efforts when needed. These preventive measures were typically long-term and routine adaptations, gradually building the defence of the natural-social system against natural disasters, thereby reducing the impact of drought on agricultural production and averting the onset of famine.

When famine did occur, preventive measures targeting food production have been ineffective. However, preventive measures for food distribution and consumption have become the material and institutional basis for emergency measures to famine relief. In ancient China, many measures were implemented to alleviate famine, among which the most commonly used were tax exemption and relief (Hao et al., 2021). From the perspective of food distribution, exemption worked by reducing the number of grain that disaster victims needed to pay to the government, thus retaining more food in the affected areas to maintain individual food supply security (Zhao, 2016). Relief, on the other hand, involved transporting grain or silver to disaster areas to balance the local food market or directly distributing to the local victims, thus maintaining the food consumption or supply security of individuals in the affected areas and alleviating famine (Zhai et al., 2020). Tax exemption and relief proved to be effective famine responses, as evidenced by records from 1587 in Pu County and Hongtong County in Shanxi province, stating "distributing grain relieved famine, saving many lives (输粟赈饥，存活良多)" and "many were saved at that time (一时多所全活)." These measures helped to alleviate food shortages in disaster-stricken areas, slow down the progression of famine, and avoid more severe social consequences, such as uprisings, that could threaten feudal rule (Bohr, 2020; Xia, 1993). Typically, it was usually after extreme droughts had caused significant damages that the state would implement exemption and relief to alleviate severe social unrest such as famine.

### 2.2.2 Data sources

The literature and records related to drought and famine in this study primarily came from two compilations, A Compendium of Chinese Meteorological Records of the Last 3000 Years (hereinafter referred to as *Compendium*) (Zhang, 2013) and the

series of books Chinese Encyclopedia of Meteorological Disasters (referred to as *Encyclopedia* hereafter) (Wen, 2008). The
two compilations have compiled a wealth of information on historical meteorological disasters and their impacts and
responses in ancient China, extracting from a variety of local chronicles and official historical records, such as Ming History
and Qing History Draft (Wang et al., 2018). The two compilations, which accurately document by years and locations, are
recognized as high-quality research materials for meteorological science and are widely used in research focused on
reconstructing and analysing extreme drought events (Zhang and Liang, 2010). From the above compiled data, drought and
famine records were extracted based on year, province, location, textual context, and source, using county as the spatial unit
(Table 1). A total of 838 drought and famine records were extracted for the two drought events, and some of the records
contained information on both drought and famine, such as "No rain in spring and summer, populace subsisting on grass and
tree bark." It should be noted that only drought-related famines were extracted, such as "a severe drought with no rain for
several months, the ground is cracked and the grass withered, and people eat each other." Famines triggered by floods, locust
plagues, and so on, were not extracted.

**Table 1. Example of drought and famine records during the Chenghua Drought**

| Year/CE | Province | County | Textual context | Data source |
|---|---|---|---|---|
| 1483 | Hebei | Huailai | No snow in winter, great drought | *Compendium* |
| 1484 | Shanxi | Dai | Little rain in spring and summer, populace subsisting on grass and tree bark | *Compendium* |
| 1484 | Henan | Weihui | Great drought in May | *Encyclopedia·Henan* |
| 1485 | Shanxi | Pu | Great drought, no rain for several months, cracked land, withered grass, cannibalism | *Encyclopedia·Shanxi* |

In the *Compendium*, in addition to records of drought and famine, it also records the exemption and relief taken by the local
government when the drought events occurred, which is the specific implementation of the state relief at the county level.
*Compendium* provides location and form of responses, for example, in 1484, Xiong County in Hebei Province experienced
drought again, and a granary was issued to provide relief and loans; in 1585 when Daming County experienced a great
drought, one-third of the agricultural tax was exempted. Based on above information, the number of counties that received
relief and exemption during the Chenghua Drought and Wanli Drought was counted.

### 2.3 Research framework and methods

#### 2.3.1 Classification of drought grades

The classification of drought grade is an important quantitative indicator for assessing the degree of drought in a specific
county and serves as a crucial factor for comparing drought severity across different counties. In the researches conducted by
numerous scholars, the duration, timing, spatial extent, and severity of drought events as recorded in literature are key
considerations for drought classification (Han et al., 2019; Zheng et al., 2014b). Ancient China was an agrarian society, and
historical records of drought typically focused on describing the impact and perception of precipitation on agricultural

production, specifically highlighting agricultural drought. Furthermore, crop yield played a vital intermediary role between natural factors such as climate and disasters, and famine (Xiao, 2020). Therefore, this study emphasized the classification of drought based on its impact on crop production. Considering that the northern region of China was primarily characterized by rainfed agriculture, the cultivation of autumn crops held a dominant position, with approximately 70% of the annual agricultural tax revenue derived from these crops (Li, 2014). Given that the spatiotemporal variability of precipitation in this area, the period from May to September represented a critical growth phase for crops, as it served as a crucial period for water supply. The amount of precipitation during this period directly affected the yield of crops (Meng, 2022) and had a greater impact on food production security in the northern China. Therefore, droughts occurring in the rainy season (May- September) were graded as Grade 1. Meanwhile, referring to the classification criteria of Yearly charts of dryness/wetness in China for the last 500-year period (CMA, 1981), the longer the drought duration and the more severe the drought in the rainy season, the higher the drought grade. Based on the above criteria, the textual information about drought was categorized into three grades (Tian et al., 2024), as shown in Table 2.

**Table 2. Classification criteria and example of drought grade** (CMA, 1981; Tian et al., 2024)

| Drought Grade | Classification Criteria | Example |
|---|---|---|
| **1**<br>**(Mild drought)** | Drought during non-rainy season | Spring drought; winter without snow |
| **2**<br>**(Moderate drought)** | Drought lasting for a season or a month during the rainy season | Spring and summer drought; summer drought; autumn drought; drought; no rain/ great drought in May |
| **3**<br>**(Severe drought)** | Severe drought lasting several months or seasons during the rainy season | Drought during summer and autumn; great summer drought; no rain from March to July |

### 2.3.2 Classification of famine grades

Famine classification is an important means of evaluating the severity of famine across different regions. To compare the severity of famine in drought-affected counties, the extracted famine records are categorized into different grades. The most commonly used indicators for assessing regional food security and famine severity are mortality rate and nutritional status (FSIN, 2022). However, in ancient Chinese literature, descriptions of famines often pertain to the contextual portrayal of the famine scene, lacking quantitative information. Nevertheless, these descriptions use diverse vocabulary, symbolizing differences about famine state, such as "it is not easy for the populace to make ends meet (民食艰难)", which means that the local residents were already suffering from the shortage of food, but they could still maintain the basic survival; "Eating grass roots, wood bark, guanyin tu (观音土)[2]" and some other terms indicated that local residents had relied on other

---

[2] Guanyin tu (观音土) is a kind of clay with fine texture that can be fatal if consumed in excess.

alternative food to survive. What's worse, records such as cannibalism, indicated that there was an extreme shortage of food, and that morality, ethics and social order had collapsed.

Thus, famine records are classified into grades based on semantic variations and the classification criteria are presented in Table 3 (Xiao, 2020; Wei, 2020; Tian et al., 2022, 2024). The criteria have been used to reconstruct the famine sequences in North China and Jiangsu-Shanghai region during the Qing Dynasty (Xiao, 2020; Wei, 2020) and have been validated. Additionally, following the characteristic of ancient Chinese literature where "unusual events are recorded, but routine events are not", if there are no famine records in the drought-affected county for a particular year, it is assumed that no famine occurred in that county during that year, and the famine grade is recorded as 0. According to Table 3, famine records from all drought-stricken counties in the study area from 1483 to 1486 and from 1585 to 1588 were classified.

**Table 3. Classification criteria and example of famine grade**

| Famine Grade | Example | Classification Criteria |
|---|---|---|
| 0 (No famine) | No famine data were recorded | |
| 1 (Mild famine) | Famine (饥), it is not easy for the populace to make ends meet (民食艰难), sporadic migration (路有流民), relief (money or grain) (赈), etc. | Severe shortage of food; can be sustained through rationing or receiving relief aid; sporadic population migration |
| 2 (Moderate famine) | Severe famine (大饥), eating grass roots and wood bark, Guanyin tu (食草根木皮、观音土), widespread migration(民流移), occasional deaths (间有死者), etc. | Extremely scarce of food; reliance on alternative food; widespread migration and displacement; occasional deaths and increasing social disorder |
| 3 (Severe famine) | Cannibalism (民相食), countless starvation deaths (死者甚众), bodies lay strewn across the ground (死者枕藉), etc. | Extreme food shortage; massive loss of life; collapse of moral ethics and societal order |

### 2.3.3 Research framework

Drawing on famine response model in ancient China and data on droughts, famines, exemption and relief, drought-affected counties could be classified into three categories (Figure 2).

Type A: Counties that experienced famine during the Chenghua Drought or Wanli Drought and received exemption or relief.

Type B: Counties that experienced famine during the drought but did not receive exemption or relief.

Type C: Counties that did not experience famine during the drought.

The distinction between Type A and Type B counties lies in whether they received government relief. Therefore, by comparing the famine severity induced by droughts of Type A and Type B, we quantitatively evaluated the effectiveness of state relief efforts in ancient China. Additionally, comparing famine severity induced by droughts within the two events of Type A analysed the changes in state emergency response capacity between the two events (Figure 2). Type C can reflect the resistance of the natural-social system to the impact of natural disaster, which means that the affected counties have

successfully blocked the transformation from drought to famine. Therefore, by calculating the proportion of counties classified as Type C during the two drought events, as well as instances of non-famine across different drought grades, a comparative analysis of the differences in defensive ability between the two events was made by this study. The calculation for famine severity caused by droughts and drought grades were detailed as follows.

245 Based on Sections 2.2.3.1 and 2.2.3.2, the drought grade ($D$) and famine grade ($F$) for each drought-affected county of the two drought events could be calculated (Eq.1, 2). From this, the average drought grade ($AD_j$) and average famine grade ($AF_j$) across the three scenarios of the two drought events could be derived (Eq.3, 4). Additionally, an $FI_j$ index was constructed to assess the severity of famine induced by drought, facilitating a comparison of famine-inducing differences across scenarios (Eq.5). It should be noted that $FI_j$ was constructed to assess the relative severity of drought-induced famine and did not

250 reflect the actual severity.

$$D = {\sum_{i=1}^{4} d_i}\big/{4} \tag{1}$$

$$F = {\sum_{i=1}^{4} f_i}\big/{4} \tag{2}$$

$$AD_j = {\sum_{n=1}^{N_j} D_n}\big/{N_j} \tag{3}$$

$$AF_j = {\sum_{n=1}^{N_j} F_n}\big/{N_j} \tag{4}$$

$$FI_j = {AF_j}\big/{AD_j} \tag{5}$$

Where: $i$ is the $i$-th year of the drought events; 4 refers to the four-year duration of the two drought events; $d$ refers to the drought grade for a given year; $f$ refers to the famine grade. $j$ stands for scenario Type $j$ ($j$ = a, b, c); $AD_j$ represents the average drought grade of Type $j$; $AF_j$ represents the average famine grade of Type $j$. $FI_j$ represents the famine-inducing

255 intensity of Type $j$; $n$ is the $n$-th drought-affected county. $N_j$ is the total number of drought-affected counties of Type $j$.

## 3 Results

### 3.1 Spatial distribution and impacts of drought and famine of the two drought events

Following the research methodology described in Section 2.3, the D and F for each drought-affected county during the

260 Chenghua Drought and Wanli Drought could be calculated. And the spatial distribution of drought and famine grades was reconstructed (Figure 3). During the Chenghua Drought, a total of 187 counties were affected, while during the Wanli

Drought, the number of affected counties reached 225, indicating that the scope of Wanli Drought was more extensive. In terms of the distribution of drought (Figure 3), it could be observed that the high-value points of drought for both drought events were concentrated in the North China Plain. Specifically, during the Chenghua Drought, the high-value drought areas were scattered, with serious drought occurring in large areas of western Shandong, southwestern Shanxi, and the vicinity of central Shaanxi. While during the Wanli Drought, the high-value drought areas were more extensive, encompassing the southern part of Hebei, western Shandong, northern Henan, and southwestern Shanxi. According to historical records, during the Chenghua Drought, many areas in the study region experienced consecutive droughts in summer and autumn, with two consecutive growing seasons of crops without sufficient rainfall, severely affecting food harvest. Some areas in western Shandong and southwestern Shanxi experienced two consecutive years of autumn drought or summer and autumn drought, resulting in population displacement. Unlike the Chenghua Drought, during the Wanli Drought, the study region experienced consecutive droughts in spring and summer. What's worse, such as western Shandong and southwestern Shanxi, there were continuous spring and summer droughts or summer droughts lasting for three years without rainfall for up to five months, leading to a scarcity of food resources. The northern part of Henan experienced consecutive spring and summer droughts or a great summer drought for four years, causing the collapse of agricultural activities.

The number of counties suffering famine in the two drought events was similar: 98 counties during the Chenghua Drought and 103 counties during the Wanli Drought. However, there were some differences. During the Wanli Drought, the high-value points of famine were located in the southern part of Shanxi, with sporadic occurrences in the central Shaanxi and northern Henan regions. In contrast, during the Chenghua Drought, the high-value famine points were more widely distributed, encompassing not only the southern part of Shanxi but also most parts of the central Shaanxi and northern Henan regions, where severe famine occurred. During the two drought events, sustained severe droughts inflicted critical damage on food production, leading to complete crop failures in many areas of Shanxi, Henan, and Guanzhong, causing severe supply-demand imbalances, market collapses, and skyrocketing rice prices. In 1484, rice prices in various parts of Henan surged from "100 Qian (a unit of currency in ancient China) per Dou (a volume unit in ancient China)" to "300 Qian per Dou". By 1486, records in Shaanxi indicated rice prices of "10,000 Qian per Dou." In contrast, during the Wanli Drought, rice prices were high but relatively more controlled. From 1586 to 1588, rice prices in many parts of Shanxi remained around "150 Qian per Dou." In 1587, many areas in Shaanxi reported rice prices of "1000 Qian per Dou," with no records of prices reaching "10,000 Qian." Food production and consumption safety could not be ensured, forcing the populace to resort to consuming tree bark and resin to stave off hunger. As alternative food sources were exhausted, massive numbers of people starved to death, leading to extreme breakdowns of social and moral norms, including instances of cannibalism.

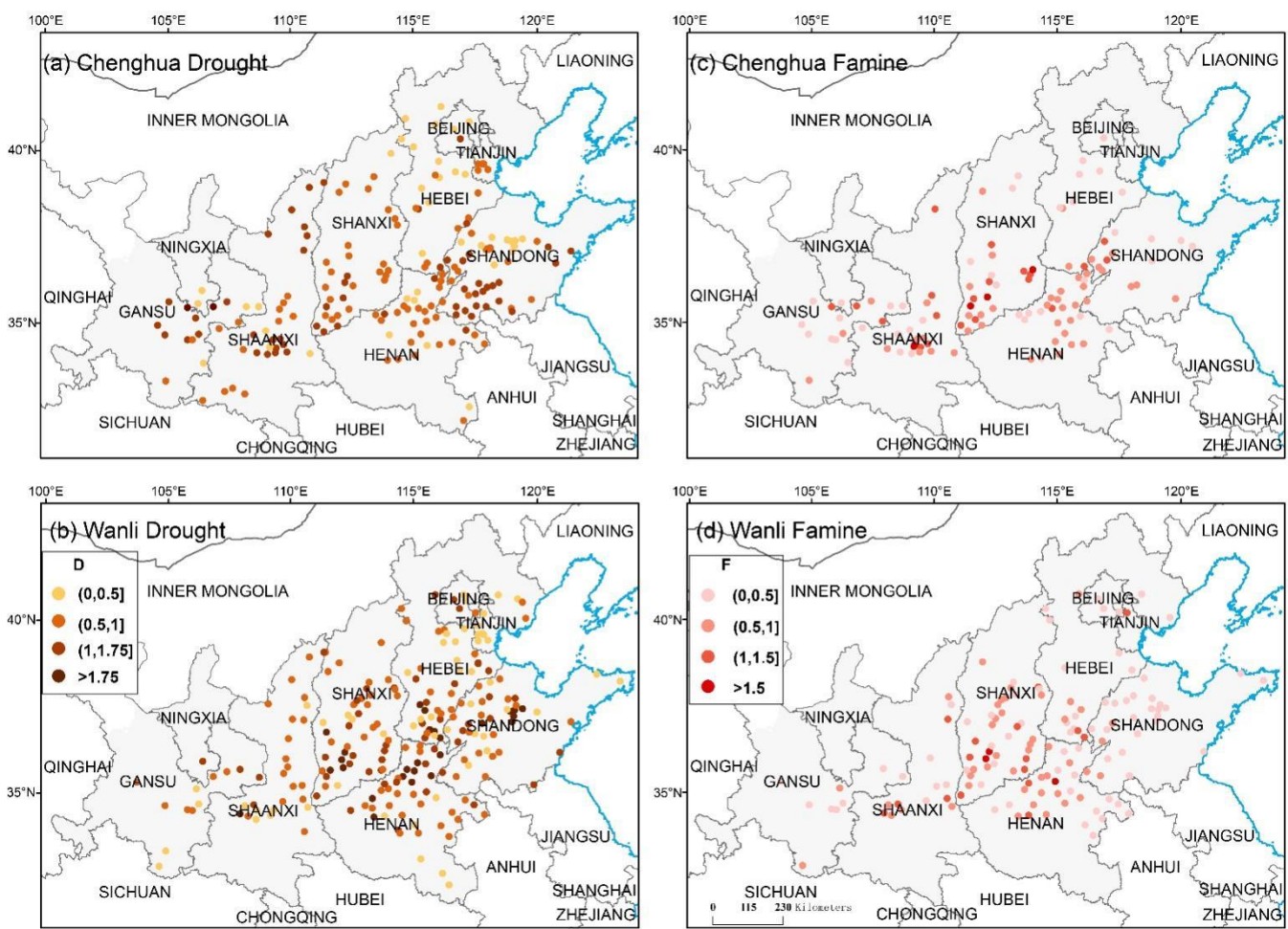

**Figure 3. Spatial distribution of drought and famine during the two drought events**

Where: (a) spatial distribution of drought grade during the Chenghua Drought; (b) spatial distribution of drought grade during the Wanli Drought; (c) spatial distribution of famine grade during the Chenghua Drought; (d) spatial distribution of famine grade during the Wanli Drought. The darker the colours, the higher the drought (*D*) and famine (*F*) grades for drought-affected counties during the two drought events.

## 3.2 Changes and causes of famine defence during the two drought events

Referring the figure 2, drought-affected counties that did not receive state relief, that is Type C and Type B, were analyzed to explore changes in the state famine resistance. According to the research method mentioned in 2.3.3, AD, AF, FI under the three scenarios of two drought events could be calculated (Table 4). As shown in Table 4, the AD for all three types of Wanli Drought were higher than those during the Chenghua Drought, indicating that Wanli Drought was more severe. However,

the proportion of Type C during the Wanli Drought (53%) was higher than that of Chenghua Drought (48%), suggesting that despite facing with more severe drought, more counties during the Wanli Drought did not experience famine. Additionally, by exploring the proportion of famine-free counties under different drought grades for both drought events (Table 5), it was evident that, generally, higher drought grade with a lower proportion of famine-free counties. Compared to the Chenghua Drought, except for drought grades of 0.25 and 1.25, the Wanli Drought consistently showed a higher proportion of famine-free counties across other drought grades. This indicated that during the Wanli Drought, the state exhibited a stronger capacity to resist drought from escalating into famine.

**Table 4: AD, AF, FI of the three types**

|  | Type C | | Type B | | | | Type A | | | |
|---|---|---|---|---|---|---|---|---|---|---|
|  | % | $AD_c$ | % | $AD_b$ | $AF_b$ | $FI_b$ | % | $AD_a$ | $AF_a$ | $FI_a$ |
| Chenghua Drought | 48% | 0.79 | 45% | 0.99 | 0.70 | 71% | 7% | 1.04 | 0.69 | 67% |
| Wanli Drought | 53% | 0.86 | 33% | 1.20 | 0.61 | 51% | 14% | 1.23 | 0.61 | 49% |

**Table 5: Proportion of famine-free counties under different drought grades ($D$)**

| $D$ | 0.25 | 0.5 | 0.75 | 1 | 1.25 | 1.5 | >1.5 |
|---|---|---|---|---|---|---|---|
| Chenghua Drought | 100% | 63% | 43% | 33% | 57% | 15% | 0 |
| Wanli Drought | 63% | 73% | 65% | 50% | 22% | 29% | 35% |

At the same time, during the Wanli Drought, the AF and FI for Type B were lower than those during the Chenghua Drought, further indicating that even when famine occurred, its severity was milder during the Wanli Drought. In terms of FI distribution of drought-affected counties for Type B (Figure 4) showed that the lower quartile, median and mean during the Wanli Drought were lower than those of Chenghua Drought. This underscored that the severity of famine during the Wanli Drought was notably less severe than during the Chenghua Drought. This suggested that, despite the absence of state relief, the protective measures implemented during the Wanli Drought effectively reduced the severity of famine, demonstrating a stronger capacity for famine defence.

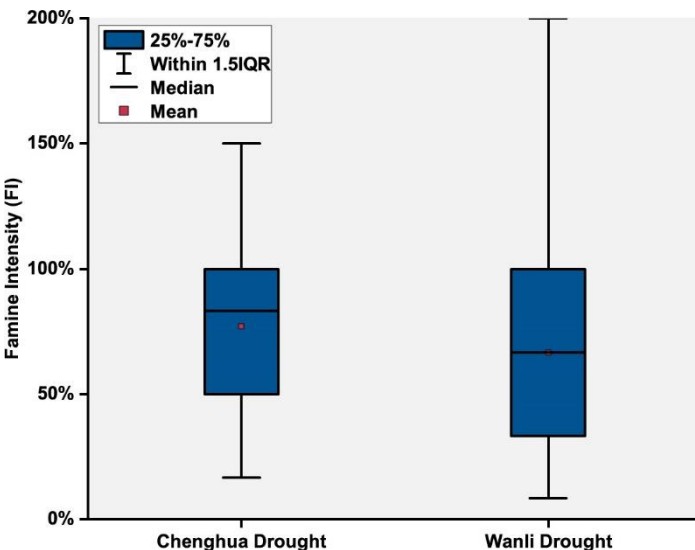

**Figure 4: The distribution of FI of drought-affected counties for Type B**

The improvement of state famine defence is closely linked to advancements in agricultural production. The advancement is
primarily manifested in land reclamation. During the Ming Dynasty, considerable efforts were made to reclaim land in order
to maintain social stability and deal with land-population conflicts. By 1583, the area of cultivated land had doubled
compared to 1391, and the per capita arable land area had nearly doubled as well (Guan and Li, 2010; Zhang, 2005). In the
research area, Henan experienced a nearly twofold increase in land area from 1491 to 1578, with significant development in
previously uncultivated mountainous regions (Tian et al., 2022). By the Wanli period, the amount of arable land in Luonan
and Shanyang counties in southern Shaanxi had grown to 2.7 times and 4.8 times of that in Jiajing period (Lv, 1996). That
implied a significant advance in agricultural production in southern Shaanxi during the Ming dynasty. From the late 14th
century to the end of the 16th century, the North China Plain, after nearly two centuries of intensive development, reemerged
as a major agricultural base in China (Cong, 1986). The increase in land reclamation and use significantly bolstered both
individual and state food reserves, playing a crucial role in preventing famine

**3.3 Changes and causes of famine emergency response during the two drought events**

Table 4 showed that, for both the Chenghua and Wanli Droughts, the $AD_a$ were slightly higher than $AD_b$, yet the $FI_a$ was
lower than $FI_b$. This suggested that state emergency measures, such as exemption and relief, effectively mitigate famine. The
proportion of Type A during the Chenghua Drought was lower compared to the Wanli Drought, indicating the scale of state
relief during the Chenghua Drought was less extensive. Data extracted from the *Compendium* revealed that the number of
counties receiving exemption and relief during the Wanli Drought was 2.5 times and 2.8 times that of the Chenghua Drought,
respectively. This reflected a significant increase in the scope of state relief. Moreover, the strengthening of state relief also
had a marked impact: the FI in Type A during the Wanli Drought was less than during the Chenghua Drought. This

demonstrated that the famine emergency responses during the Wanli Drought were more effective than those during the Chenghua Drought.

The increased intensity of state relief preserved and delivered more food to disaster-stricken areas, which helped to alleviate the hardships and burdens faced by the peasants. Intensity of the emergency responses was closely related to the financial resources and economic conditions at that time (Chen, 2015). If the finance was strained and the economy was in a slump, the priority of famine relief measures naturally competed to other national affairs (Will, 2006). Wanli Drought coincided with the aftermath of reforms implemented by Zhang Juzheng, which significantly enhanced the national finances.

Consequently, the tax grain in the Jing and Tong Granaries substantially increased, leading to abundant food supplies, with the Jing granary once holding as much as 18.18 million Dan of grain. Starting from the middle of the Ming Dynasty, the Taicang Treasury gradually evolved into a central core financial institution responsible for important national financial expenditures such as northern border provisions, national relief, and significant financial levies (Su, 2010). Its income and expenditure closely related to the national finances. The increase or decrease in its income reflects the social and economic

dynamics at that time and the position of silver in the circulating currency (Quan, 2011). In terms of silver income in the Taicang Treasury after the middle of the Ming Dynasty, it significantly increased from the Mid-16th century onwards (Figure 5). On one hand, this was related to the promotion of the silver currency at that time. On the other hand, it was related to the vigorous development of national salt policies and the commodity economy. Meanwhile, scholars have compiled the Ming Dynasty's fiscal revenue situation from the 15th to 17th century (Liu, 2012). Rice and wheat were major

sources of fiscal revenue, which exhibited an upward trend from the 15th to 17th century. Overall, the total revenue after converting to silver showed a clear upward trend. Therefore, it can be inferred that the development of the national economy provided the material basis for relief and exemption measures. With the increase in national fiscal revenue and economic development, the ability to provide exemptions and relief also grew stronger. The significant magnitude of exemption and relief allowed for the retention and distribution of more relief grains in disaster-hit areas during the Wanli Drought, thereby

mitigating the development of famine.

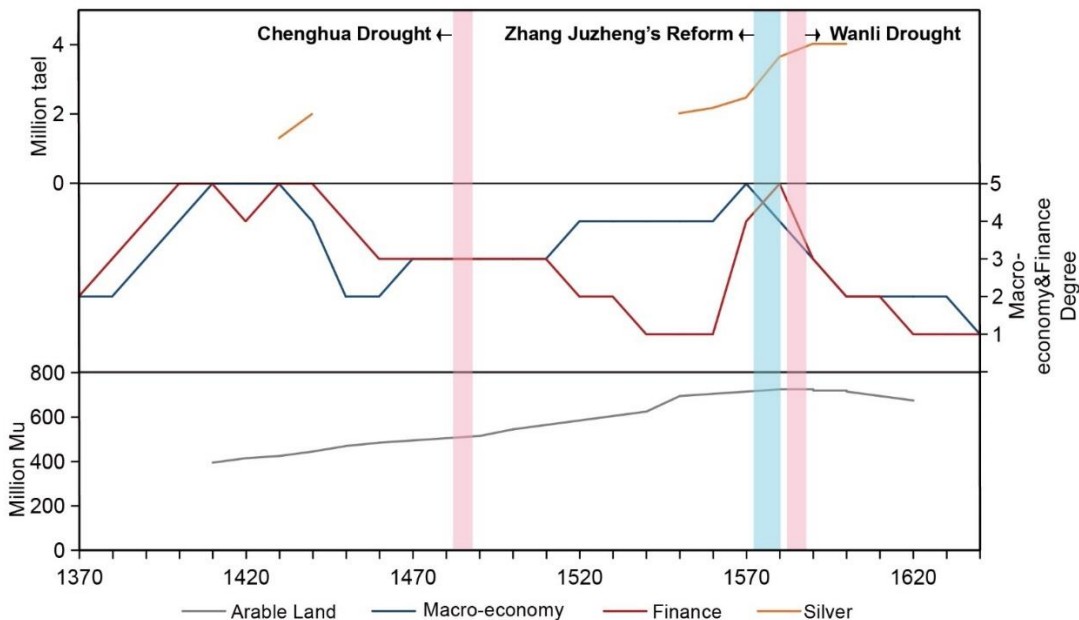

**Figure 5. Change of arable land** (Guan and Li, 2010)**, finance** (Wei et al., 2014)**, macro-economy** (Wei et al., 2015) **and silver** (Su, 2010) **during the Ming Dynasty**

**4 Discussion**

**4.1 Regional differences in response capacity**

When comparing the impacts and responses of the two drought events, it was evident that there were differences in the impacts and responses of drought among different provinces. Using the provincial boundaries of the Ming Dynasty as a reference (Han and Yang, 2021), Beijing, Tianjin, and Hebei were combined into the B/T/H. This area was the political

centre of the state, characterized by advanced agricultural practices and dense populations. Shaanxi, Ningxia, and parts of Gansu were grouped as the S/G/N, all located in the Ming-era Shaanxi Province, a frontier region at that time. Although Henan, Shandong, and Shanxi were all situated in northern China, they displayed significant differences in their ecological environments and economic development. According to AD, AF, and FI (Figure 6 and Table 6), the study regions can be categorized into three groups:

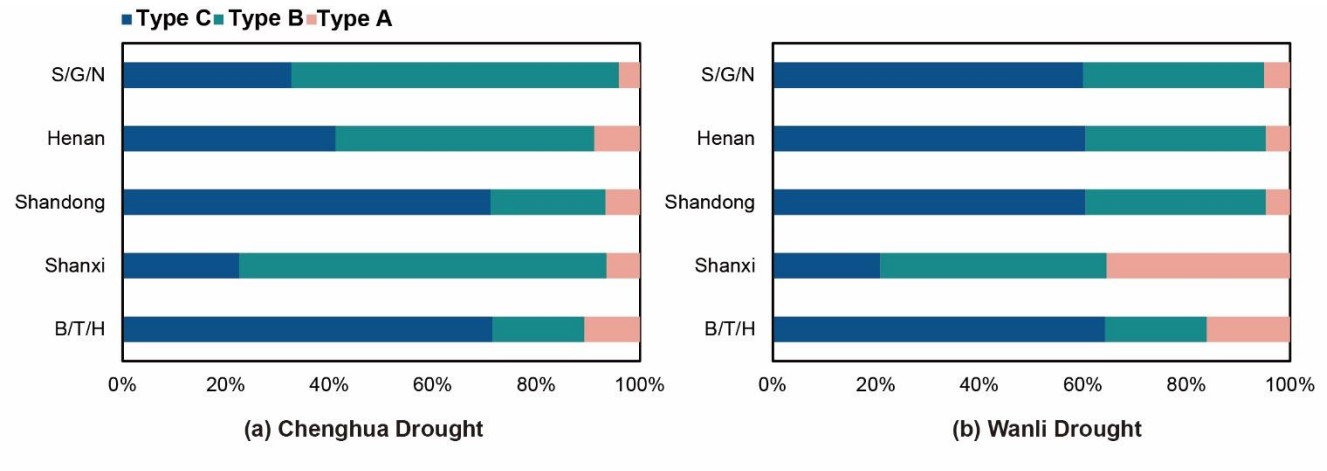

**Figure 6: The proportion of drought-affected counties categorized into three types (Type A, B, C) in each region during the two extreme drought events**

**Table 6: AD, AF, FI in different regions of the two drought events**

| | Chenghua Drought | | | | | Wanli Drought | | | |
| | | Type A & Type B | | | | | Type A & Type B | | |
| | AD | $AD_{ab}$ | $AF_{ab}$ | $FI_{ab}$ | | AD | $AD_{ab}$ | $AF_{ab}$ | $F_{ab}$ |
|---|---|---|---|---|---|---|---|---|---|
| **B/T/H** | 0.6 | 0.8 | 0.6 | 73% | | 0.9 | 1.3 | 0.6 | 46% |
| **Shandong** | 1.0 | 1.0 | 0.7 | 63% | | 1.1 | 1.5 | 0.4 | 30% |
| **Shanxi** | 0.9 | 1.0 | 0.8 | 82% | | 1.1 | 1.1 | 0.8 | 66% |
| **Henan** | 0.8 | 0.8 | 0.7 | 82% | | 1.2 | 1.4 | 0.7 | 46% |
| **S/G/N** | 1.1 | 1.2 | 0.7 | 61% | | 0.8 | 0.9 | 0.6 | 65% |

The first category included the B/T/H and Shandong. In both regions, Type C dominated. In the Chenghua and Wanli Droughts, Type C consistently accounted for over 60% of the total (Figure 6). This meaned that even during droughts, even severe droughts (such as Shandong with AD of 1.0 and 1.1), more than half of the affected areas did not experience famine. Moreover, famine intensity in both regions where famine did occur (Type A and Type B) showed that the degree of famine was relatively low, with FI in the Wanli drought being just 46% and 30%, respectively. These findings indicated strong famine response capabilities in the two regions. As the political centre of the Ming Dynasty, it was critical for maintaining B/T/H food security. Therefore, during droughts, this region received a substantial share of state relief efforts. Table 6 and Table 7 revealed that even with moderate drought (AD of 0.6 and 0.9), more than 20% of the total relief efforts were concentrated in this area, reflecting a significant allocation of national resources. Additionally, both the B/T/H and Shandong housed major granaries, which bolstered their grain reserves. During the Wanli drought, approximately 88% of the relief grain came from the Linqing and Dezhou granaries in Shandong province (Tian et al., 2024).

**Table 7: Proportion of state effort across different regions**

|                    | B/T/H | Shanxi | Shandong | Henan | S/G/N |
|--------------------|-------|--------|----------|-------|-------|
| **Chenghua Drought** | 23%   | 15%    | 23%      | 23%   | 15%   |
| **Wanli Drought**   | 35%   | 48%    | 9%       | 4%    | 4%    |

The second category was Shanxi. In Shanxi, the majority of counties in Shanxi fell under Type B and Type A, accounting for nearly 80% of all counties affected by the two drought events. Particularly during the Chenghua Drought, Type B made up 70% of the total, highlighting limited ability to prevent drought from escalating into famine in Shanxi. Moreover, Shanxi consistently exhibited the highest famine severity across all provinces during both droughts. Even though the AD in Shanxi were similar to those in Shandong, AF was significantly higher. Despite receiving substantial relief during the Wanli Drought, Shanxi still experienced the most severe famine, underscoring its weak resilience. This vulnerability could be attributed to severe land pressure and vulnerable ecological environment. During the Ming dynasty, the average land area per capita in Shanxi was just 6.51 Mu[3] per person, much lower than the national average of 11.56 Mu per person, indicating significant population-land conflict (Liang, 2008). Additionally, over 80% of soil in Shanxi consisted of loess and secondary loess, which were prone to erosion. Moreover, Shanxi was predominantly mountainous with limited plains, making its ecological environment vulnerable, which posed significant challenges to grain production (Shuang and Wang, 2020). As a result of agricultural damage, Shanxi also had the highest number of migrants among the provinces in the study area (Li, 1998).

The third category included Henan and the S/G/N. In both regions, the distribution of the three types shifted significantly: the proportion of Type C increased, while that of Type B decreased. This indicated that many drought-affected counties in these regions had greatly improved their ability to prevent drought from evolving into famine. Although the FI remained high, especially in the S/G/N where it exceeded 60% during both drought events, Henan experienced a notable reduction in FI, dropping to 46% during the Wanli drought. As discussed in Section 3.2, the expansion of cultivated land and the introduction of American crops during the Ming dynasty played a crucial role in enhancing famine resilience in both regions. Further research and historical documentation are required to explore additional factors contributing to these changes.

**4.2 Inheritance and development of historical disaster experience**

Based on the above section, it is evident that historical response capacity varied across both spatial and temporal scales. From Chenghua Drought to Wanli Drought—a span of one hundred years—the defensive ability and emergency response capabilities of the state both improved. Nonetheless, during the Wanli Drought, large numbers of people were still displaced, and in 32 counties, famine led to cannibalism within the study area. From the Western Han (202 BC- 8 CE) to the Qing Dynasty (1644-1911 CE), famine persisted as an inescapable consequence of climate shocks, with cyclical fluctuations in

---

[3] Mu is a unit of area. 1Mu ≈0.067 hectares.

severity (Fang et al., 2015) (Figure 7c). However, in the modern era, China has markedly strengthened its capacity to manage climate shocks, achieving significant progress in eradicating hunger (World Bank, 2024) (Figure 7d), a success that is closely tied to the establishment of a modern disaster management system. Compared to historical periods, this system has inherited and developed a wealth of ancient knowledge and insights.

From an inheritance perspective, throughout history and into the present, Chinese have consistently emphasized the development of agricultural technology and improvement of land productivity to bolster the defensive ability of the natural-social system. Irrigation projects illustrate this continuity. According to the International Commission on Irrigation and Drainage (World Heritage Irrigation Structures, 2024), as of 2024, China possesses 38 World Irrigation Heritage sites—the most diverse, widespread, and beneficial irrigation heritage of any country. By 2023, effective irrigated land area in China

had exceeded 700 million hectares (National Bureau of Statistics of China, 2024). In addition to sustained attention to agricultural technologies, the government has consistently played a central role in disaster management, both in historical and modern society. When confronted with severe natural disasters, the government has repeatedly demonstrated robust capacities for organizing and mobilizing social resources, ensuring that relief efforts are conducted in an organized and orderly way. The government thus remains the leading force in disaster response. Furthermore, influenced by Confucian

culture, society has developed a "help from all sides when one area is in difficulty" ethos. This cultural ethos not only mobilizes substantial human and material resources but also build a shared understanding of prioritizing disaster relief (Shi and Zhang, 2013). In modern times, nationwide response to major disasters has been formed.

In terms of development, scientific and technological advancements stemming from the Industrial Revolution have driven substantial progress in agricultural technology, significantly boosting land productivity and resistance (Figure 7a). From the

Han dynasty to the end of the Qing dynasty, increases in land yield per-hectare did not exceed 1,417 kg/ha (Wu, 2007). However, from 1978 to 2023, grain yield per hectare rose from 2,526 kg/ha to 5,845 kg/ha (National Bureau of Statistics of China, 2024), significantly enhancing land-use efficiency. Additionally, over time, China has transitioned from an agrarian to an industrial society, improving its response capacity to a new level (Figure 7b). Agricultural losses caused by climate shock no longer disrupted heavily food supply and consumption security. During this transformation, societal response

strategies have changed significantly. In historical periods, the state collected substantial grain through land taxes on peasants, then redistributed this grain through storage and relief to ensure food security during famine years. Grain circulation through market exchange was often limited, restricting access to food for poor person. Today, a grain circulation system has been established, based on central and local reserves and supported by a diverse mix of private, state-owned, and mixed-ownership entities, significantly enhancing social grain circulation capacity and food availability for individuals.

Consequently, modern China has effectively reduced the likelihood of famine (Figure 7d) and ensured food security.

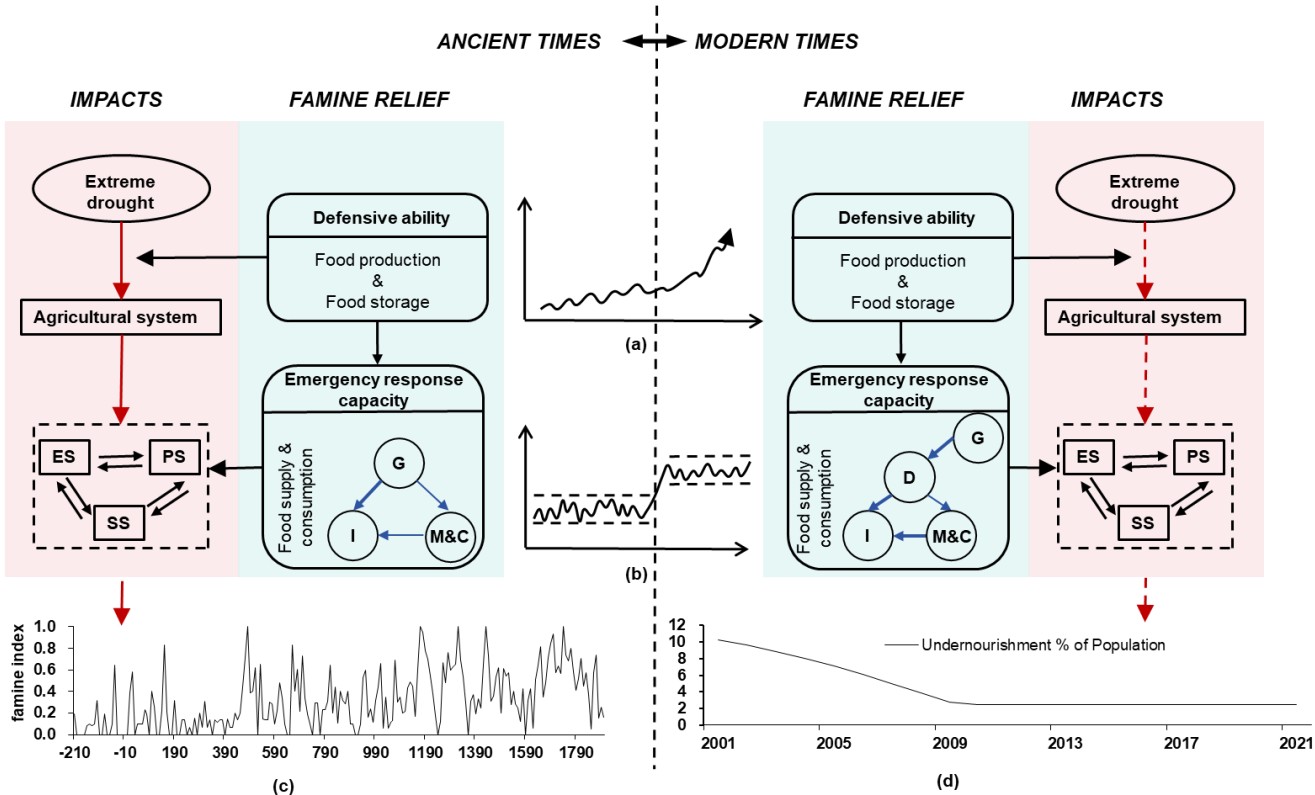

**Figure 7: The inheritance and development of famine relief from ancient to modern times**

Where: Red solid line: strong impact; red dashed line: weak impact; ES: economic system; PS: population system; SS: social system; G: governments; M&C: market and clarity; I: individuals; Blue thick line: strong relationship; blue thin line: weak relationship; (a) change of defensive ability; (b) change of emergency response capacity; (c) famine index (210 BC-1900 CE); (d) undernourishment % of population

## 5 Main conclusion

This study developed a model of extreme drought-induced famine processes and response mechanisms in ancient China, drawing on historical records and research findings. Spatial distribution of drought and famine during the Chenghua Drought and the Wanli Drought was constructed. By categorizing drought-affected counties into three types, a comparative analysis of the differences in famine severity and response effectiveness between the Chenghua and Wanli droughts was conducted. The study also tried to explain changes in response ability based on historical sources. The key findings were as follows:

(1) Comparison of Drought Events: Although the Wanli Drought presented more severe than the Chenghua Drought, the famine it caused was less severe, with a lower famine intensity. The proportion of Type C during the Wanli Drought was higher than during the Chenghua Drought, while the AD was also higher. This indicated that society demonstrated stronger defence during the Wanli Drought, preventing many drought-affected counties from descending into famine. Furthermore, in both drought events, the FI in Type B was higher than in Type A, suggesting that state relief were effective. The increased intensity of state relief during the Wanli Drought further contributed to its lower famine severity.

(2) Factors Influencing Response Capacity: Factors such as the expansion of cultivated land, and the economic and fiscal health of state all influenced societal response capacity. During the Wanli Drought, the expansion of arable land, sound economic conditions, and the trend toward silver monetization all played positive roles in improving famine preparedness and relief.

(3) Regional Variations in Response Capacity: Significant regional differences in response capacity were evident. Shandong and the B/T/H (Beijing, Tianjin, and Hebei) displayed strong resistance to famine, with the majority of drought-affected areas in these regions avoiding famine, likely due to their political significance and substantial grain reserves. In contrast, Shanxi remained highly vulnerable to famine, likely due to its vulnerable ecological environment. From Chenghua Drought to Wanli Drought, the resistance of Henan and S/G/N (Shaanxi, Gansu, and Ningxia) to drought improved significantly. The causes of this change require further investigation.

Additionally, this study explores how China's modern disaster management system has inherited and developed upon ancient famine relief practices. This research highlights many areas that warrant further exploration, such as the relationship between individual and state efforts in famine preparedness and relief. In Ming China, an agrarian society, the state's grain taxes were primarily drawn from land production. After paying taxes and setting aside enough for daily subsistence, peasants could continue agricultural production. The state, in turn, used these tax revenues for famine preparedness and relief. The complex interactions between these efforts need deeper exploration.

**Data availability.** The data that supporting the findings of this study are available on request from authors.

**Author contributions.** Fangyu Tian developed the research method, collected drought, famine and state relief data, and drafted the manuscript. Yun Su supervised the research of this study and developed the research method. Xudong Chen and Le Tao reviewed the manuscript and revised the data.

**Competing interests.** The authors declare that they have no conflict of interest.

**Acknowledgements**. Supported by the National Key Research and Development Program of China [2018YFA0605602] and the Major Program of National Fund of Philosophy and Social Science of China [grant number: 22&ZD223].

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
