# Peer review of "Enhancement of state response capability and famine mitigation: A comparative analysis of two drought events in northern China during the Ming Dynasty"

_Natural Hazards and Earth System Sciences, 2024_

## Author Comment (AC1)

We sincerely thank Momin, Samar for your constructive and kind word on our manuscript. Our responses are divided into two sections: responses to specific comments and responses to technical comments.

**Responses to specific comments:**

*Historical Vs modern climate change adaption:*

Thank you for your suggestion, which has encouraged us to emphasize the connection between historical experiences and the present. There are similarities between modern climate change adaptation in China and historical disaster responses regarding actors, processes, and measures, even though the productivity of agricultural societies differs significantly from that of contemporary society. Notably, modern disaster management still reflects traditional characteristics, particularly "government-led, collectivist". Therefore, we will add a brief discussion, Section 4.2 to explore the differences and inheritance between modern disaster management in China and historical disaster responses.

*Formulations:*

Thank you for your suggestion about formulations.

We also noticed some confusion regarding indicators such as AD, AD1, AF, and FI in Table 4, Table 6, as well as in lines 285 and 249. Therefore, we will clarify these distinctions during the revision process.

In our study, AD referred to the average drought degree over four years in drought-affected counties under different scenarios (Type A, Type B, Type C); AF referred to the average famine degree in drought-affected counties during the four-year drought events under different scenarios. The ratio AF/AD was considered a measure of the severity of drought-induced famine, used to compare how much droughts under different scenarios and events translate into famines. We will provide further explanations of these terms in the manuscript.

*Graphical summaries:*

During the revision process, we found data from published literature (Guan and Li, 2010; Yin et al, 2016; Chen et al, 2024) that represent the macro-economy, fiscal degree, and arable land area during the Ming Dynasty. We will add these data with the silver in Figure 5 to jointly illustrate the social context of the two drought events. Additionally, we will provide further explanations in Sections 3.2 and 3.3.

[Figure]

Figure 5: Macro-economy, finance, arable land and silver during the Ming Dynasty.

**Responses to technical comments:**

We sincerely appreciate the reviewer's technical comments, which have helped us enhance the accuracy and clarity of our manuscript. In response to these suggestions, we will revise sentence structures, add annotations, and adjust the reference formatting accordingly.

---

## Author Comment (AC2)

We sincerely appreciate the reviewer's suggestions, which have greatly improved our article. Our responses are organized into two categories: responses to specific comments and responses to technical comments.

**Responses to specific comments:**
We appreciate the reviewer's suggestions regarding data sources. Indeed, the Ming Shilu is a valuable document for studying the imperial court's response during the drought events. We have collected data on relief efforts (赈济) from the Ming Shilu for the two drought events and conducted preliminary analyses. However, the conclusions were not relevant to the main arguments of this article, and there was insufficient historical evidence to support them further.
We also examined the records of tax exemptions for the two drought events in the Ming Shilu, but due to differences in textual record, it was not possible to compare changes in the intensity of tax exemptions between the two drought events. Therefore, we did not use data from the Ming Shilu in our study. We will clarify these in the data sources section of the manuscript.

*Below are the conclusions on relief efforts that were not presented in our manuscript and the tax exemption records:*
*During data collection, we extracted a total of 40 records regarding relief efforts from Volume 236-285 of Ming XianZong Shi Lu and Volume 157-206 of Ming ShenZong Shi Lu about Chenghua Drought (1483-1486 CE) and Wanli Drought (1585-1588 CE). Additionally, we have conducted some preliminary analyze. The results were as follows:*

The Ming Dynasty mainly took two forms of relief: grain and silver. During the two drought events, grain relief and silver relief had obvious changes in quantity. During the Chenghua Drought, relief were predominantly in the form of grain, amounting to 1.42 million Dan. In contrast, during the Wanli Drought, grain relief decreased to approximately 0.82 million dan. Silver relief, on the other hand, showed a marked increase—from 520 thousand taels during the Chenghua Drought to 1.42 million taels during the Wanli Drought.
Based on the grain price of 2 taels per Dan in famine years (Chen, 2016), we conducted a preliminary estimation of the proportion of relief silver received by each province during the two drought periods (Table 1). It was evident that the proportion of receiving silver had increased significantly. From the Chenghua Drought to the Wanli Drought, the status of silver relief became prominently prominent.

Table 1: The proportion of relief silver (converted into grain) received by the five northern provinces during the two droughts.

|                      | Beizhili | Shandong | Shanxi | Henan | Shaanxi |
|----------------------|----------|----------|--------|-------|---------|
| **Chenghua Drought** | 0%       | 4%       | 23%    | 16%   | 14%     |
| **Wanli Drought**    | 35%      | 19%      | 75%    | 18%   | 87%     |

Silver relief possessed its own advantages compared to other relief methods such as grain relief. During large-scale natural disasters, when numerous famine-stricken people required assistance, providing relief grain would necessitate a large quantity of food to be transported from areas with abundant supplies to the affected regions. This process involved immense manpower, resources, and

exorbitant transportation costs. Additionally, due to inconvenient transportation means, relying on carts, horses, and ships at that time, it took a substantial amount of time for the grain to reach the disaster-stricken areas, risking missing the optimal relief timing. In contrast, silver relief effectively addressed these challenges. Silver had higher value and smaller volume, making it easier to store, carry, and transport. By issuing silver to suitable recipients based on the severity of disaster and the actual situation of famine-stricken individuals, silver could be allocated towards those not severely affected. Those could use silver to purchase grains or other essential commodities from surrounding areas, thereby alleviating the pressure on food supply in the affected regions (Huang, 2014) and subsequently mitigating famine. However, in the case of extreme drought events, when food supplies in the affected areas were exhausted, there was often a situation where grain was unavailable for purchase despite having money. Whether silver relief could still exert its advantages in such circumstances requires further historical evidence.

*Based on the data we collected, no historical records indicate that the form of relief significantly impacted the effectiveness of famine responses during the Chenghua and Wanli droughts. Assessing the efficiency of disaster relief efforts requires more data, which we were unable to fully address in this study.*

*Secondly, when collecting data on tax exemptions, there is a significant difference between the records of exemptions during the Chenghua and Wanli reigns in Ming Shi Lu.* The records from the Chenghua period include details such as location, type of disaster, and quantities, for example,

*"Due to drought, autumn grain and grass tax of last year in Datong, Shanxi and other prefectures were exempted, amounting to more than 2.3 million Dan of grain and more than 4.34 million bundles of grass."*
*(以旱灾免山西大同等府卫去年秋粮子粒二十三万余石马草四十三万四千余束)*

In contrast, the records from the Wanli period are less specific regarding quantities, for example

*"The decree ordered that the tax could be exempted at varying levels for the disaster-stricken populations in Shaanxi and Shanxi"*
*(诏狭西山西被灾民屯钱粮蠲折有差)*

*It is quite challenging to compare the magnitude of tax exemptions between the two drought events. Therefore, for the reasons mentioned above, we have chosen not to include data from the Ming Shi Lu in this study. We will provide an explanation in the paper*

**Responses to technical comments:**
We are very grateful for your insightful technical comments, which have significantly contributed to the rigor of this manuscript. In response to your valuable feedback, we will ensure that the necessary references are added, revisions are made, and supplementary annotations are incorporated at the relevant points within the article. As for the American crops you mentioned, more researches (He, 1979; You, 1989; Wang, 2004) have led us to recognize that we had overstated their influence

on the northern regions during the Wanli Drought. As a result, we have decided to remove the section discussing American crops to maintain the accuracy and integrity of our analysis.

---

## Author Comment (AC3)

We appreciate the reviewer for the genuine and frank comments on our manuscript, which has boosted our confidence in our research. Our point-by-point responses are as followed, divided into responses to specific comments and responses to technical comments.

**Responses to specific comments:**

We sincerely thank the reviewer for helping us explore the modern meaning of historical case studies. We will add a discussion, section 4.2, to analyze the inheritance of historical disaster response in modern China.

The disaster response measures and ideas in the past have a profound influence on modern-day disaster management in China. For example, the **process of disaster management in modern time** consists of three phases: pre-disaster prevention, emergency, and post-disaster reconstruction and restoration (Patel & Hastak, 2013; Safapour et al., 2021), which is similar to the idea of famine preparedness and famine relief in ancient China. To mitigate famine and ensure food security, contemporary China still pays more attention to **similar famine mitigation measures about food production and food reserve** (Bruins & Bu, 2006), such as the construction and maintenance of water conservancy projects, crop improvement, and policies to protect arable land from being encroached upon by urban development. Regarding **leadership** in disaster management, the government always plays a central role. As for **disaster culture**, the collectivist spirit of "Support from all sides when one side is in trouble" remains a guiding principle. Therefore, we will add a discussion section to explore the impact of historical response experiences on contemporary disaster management.

**Responses to technical comments:**

We sincerely appreciate the reviewer's technical comments, which have enhanced the rigor of our manuscript and ensured consistency in formatting.

Firstly, after examing the discussion of American crops, we found    their impact on the Wanli drought was overestimated. As a result, we will remove the discussion related to American crops.

Secondly, in Table 4 and Table 6, the term "AD1" is used to represent different meanings (as indicated by the formulas below). Using the same notation in both cases may lead to confusion. We will adopt distinct symbols to clearly differentiate them.

In Table 4, AD1 represents the average drought severity of two drought events under the Type C, calculated using the following formula:

$$AD = \left. \sum_{n=1}^{N} \sum_{i=1}^{4} d \middle/ 4 \times N \right.$$

AD represents the average drought grade; 4 refers to the four-year duration of the two drought events; $i$ is the $i$-th year of the drought; $n$ is the $n$-th drought-affected country. $N$ is the total number of drought-affected counties under the Type C.

In Table 6, AD1 represents the average drought severity of two drought events under the Type A and Type B calculated using the following formula:

$$AD' = \left. \sum_{n=1}^{N} \sum_{i=1}^{4} d \middle/ 4 \times N' \right.$$

$AD'$ represents the average drought grade; 4 refers to the four-year duration of the two drought events; $i$ is the $i$-th year of the drought; $n$ is the $n$-th drought-affected country. $N$ is the total number of drought-affected counties under the Type A and Type B.

Thirdly, we will review the reference formatting in the manuscript and make corrections to any citations with formatting errors.

*Thank you very much for the reviewer's suggestions. We noticed that the same set of comments was uploaded four times. To avoid unnecessary messages, we have responded to just one. We appreciate your understanding.*

---

## Author Response (AR1)

**Dear Editor and reviewers,**

We would like to express our sincere gratitude for your suggestions regarding our manuscript. We thank the editor and reviewers for the time and effort that have put into reviewing the previous version of the manuscript. Your suggestions have enabled us to improve our work. Based on the instructions provided in the letter, we uploaded the file of the revised manuscript. Accordingly, we have uploaded a copy of the original manuscript with the changes. Appended to this letter is our point-by-point response to the comments raised by the editor and reviewers. The comments are reproduced and our **responses** are given directly afterward in a different color **(blue)** based on revised manuscript.

Thanks again!

Sincerely,

Authors

**Response to editor, Dr. Khalid,**

Clarification of the Term "Ming Dynasty": To enhance readability and comprehension for a wider audience, it is recommended that the term "Ming Dynasty" be clearly defined in the "Overview of the Study Region" section. This will provide readers with the necessary historical context to understand the period under study and the socio-economic conditions prevalent during that time.

Response: We sincerely appreciate the editor's positive feedback on our model development, research framework, methodology, and data sources, which has greatly encouraged us in revising the manuscript. In response to the term "Ming Dynasty," we have expanded the **Introduction in Line 77-82** of revised manuscript to add details on dynasty's period, climate and social context, providing readers with the necessary historical context during that time.

**Response to reviewer, Dr. Samar Momin,**

**Responses to specific comments:**

Historical Vs modern climate change adaption:

It would be useful to have a section with a simple comparison or explanation between these historical responses and modern-day disaster management strategies adopted in China. This would make the research more relevant to current discussions on climate change adaptation.

**Response:** Thank you for your suggestion, which has encouraged us to find out the connection between historical experiences and the present. We have added **Discussion 4.2: Inheritance and development of historical disaster experience (Line 413-454)**, which explores how historical experience has shaped modern climate adaptation and disaster management from the perspectives of inheritance and development. In terms of inheritance, from historical period to the present, the Chinese

people have prioritized advancements in agricultural techniques and land productivity to strengthen defense against natural disasters. And the government has consistently played a leading role in disaster management, fostering a social support mechanism — "help from all sides when one area is in difficulty" — rooted in Confucian culture.

In terms of development, scientific and technological progress has led to significant advancements in agricultural techniques, greatly enhancing land productivity and defensive ability. Additionally, China has transitioned from an agrarian society to an industrial one, resulting in substantial shifts in social response mechanisms. Traditional methods, such as grain redistribution through agricultural taxes and relief efforts, have been replaced by a modern emphasis on developing grain markets to strengthen food distribution and ensure individual food security. This shift has allowed emergency response capacities to reach a new level.

Formulations: Could the authors provide a clear explanation (in the text) about how the formulations were derived?

**Response:** Thank you for your suggestion about formulation. We have added the derivation process of the formulas in **Section 2.2.3.3, Line 234-238; Line 230-252**. Using Formula 1 and Formula 2, we can calculate the average drought degree **(D)** and average famine degree **(F)** for each drought-affected county. They are used to display the spatial distribution of drought and famine about Chenghua Drought and Wanli Drought (**Figure 3**). Based on this, we can obtain the average drought degree **(AD_j)** and average famine degree **(AF_j)** for three different scenarios of the two drought events. Additionally, we constructed the index **FI_j** to assess the severity of famine caused by drought, which is used to compare differences in famine severity across different scenarios. It should be noted that the construction of **FI_j** is intended to assess the relative severity of drought-induced famine, and cannot reflect the actual severity.

$$D = {\sum_{i=1}^{4} d_i}\Big/{4} \tag{1}$$

$$F = {\sum_{i=1}^{4} f_i}\Big/{4} \tag{2}$$

$$AD_j = {\sum_{n=1}^{N_j} D_n}\Big/{N_j} \tag{3}$$

$$AF_j = {\sum_{n=1}^{N_j} F_n}\Big/{N_j} \tag{4}$$

$$FI_j = {AF_j}\Big/{AD_j} \tag{5}$$

In the equations, $d$ represents the drought severity for a drought year; $f$ represents the famine severity for a drought year; $j$ denotes the scenario type (where $j$ = a, b, c); $AD_j$ represents the average drought intensity for type $j$; $AF_j$ represents the average famine degree for type $j$; $FI_j$ represents the famine severity caused by drought for type $j$; 4 indicates that both drought events lasted for four years; $i$ denotes the year of the $i$-th drought event; $n$ represents the $n$-th drought-affected county; and $N_j$ represents the number of drought-affected counties in type $j$.

The spatial distribution maps and graphs are useful. However, the authors could add additional visual representations of the comparisons between the two droughts (e.g., timeline diagrams) could further clarify the narrative.

**Response:** Thank you for your suggestion on graphical summaries. We also believe that graphical summaries can help us express our points more effectively. Therefore, we have collected data on fiscal degree (Wei et al., 2014), macroeconomy (Wei et al., 2015), arable land (Guan and Li, 2010), and silver quantities (Su, 2010) of Ming Dynasty to help visually compare the social contexts of two drought events (**figure 5**). Meanwhile, we have added these references in the text.

[Figure]

**Responses to technical comments:**
We sincerely appreciate the reviewer's technical comments, which have helped us enhance the accuracy and clarity of our manuscript.

In general spaces between text and in-text citation are missing throughout the manuscript.
Response: We have updated the in-text reference formatting and adjusted the reference list to ensure consistency and compliance with citation standards.

Grammatical/sentence structure:
Requires correction: "Give that famine often stems from poor harvests..."
Corrected: "Given that famine often stems from poor harvests..."
Requires correction: "...as a time when ancient famine response policies were highly well-develop in China."
Corrected: "...as a time when ancient famine response policies were highly well-developed in China."

Response: We have corrected accordingly.

Needs clarification: Line 135, Ancient China gradually developed a comprehensive famine response system that deal with each step of processes to extreme drought-induced famines (Figure 2).

Needs clarification: Line 147, In ancient China, there were various emergency measures to mitigate famine, among which exemption and relied being the most common (Hao et al., 2021).

Response: We have revision in **Line 140** and **Line 157-158.**

Throughout the manuscript counties classifications Type A, Type B and Type C are written as "Type a" or "type a", I believe that since it is a classification, the Latin letter needs to be capitalized for example use Type "A" or type "A" consistently.

Response: We have revised "type a," "type b," and "type c" to "Type A," "Type B," and "Type C." in the text and updated the Figure 2, 3, 6.

Clarification: What is the unit (mu) here? "6.51 mu per person"

Response：The unit "mu" (亩) is a unit of area, where 1 mu ≈ 0.0067 hectares. We have added a note to clarify it.

**Response to reviewer #2,**

**Responses to specific comments:**

A Compendium of Chinese Meteorological Records of the Last 3000 Years (中国三千年气象记录总集) is the main source of historical records about the government's response measures. However, the records of post-disaster response (relief, exemption) kept in this book are incomplete, as the purpose of the contributors is to collect meteorological disasters and their consequences. Secondly, most of these records are extracted from historical local Chronicles, which mainly record the response behavior of local society, while the measures from the imperial court (central government) are not comprehensive. It is suggested to supplement the response measures (especially tax exemption and grain dispatching) of the imperial court in the Ming Shilu (明实录), compilation of government archives of the Ming Dynasty), which will be more comprehensive and helpful for later comparison of the strength of government response capability in Chenghua and Wanli reigns.

Response: We appreciate the reviewer's suggestions regarding data sources. Indeed, the Ming Shilu is a valuable document for studying the imperial court's response during the drought events. However, after examining and analyzing the records in the Ming Shilu, we found significant discrepancies in the documentation of the response data for the two drought events. For instance, during the Chenghua period, records of tax exemption included details such as the location, type of disaster, and quantities:

*"Due to drought, autumn grain and grass tax of last year in Datong, Shanxi and other prefectures were exempted, amounting to more than 2.3 million Dan of grain and more than 4.34 million bundles of grass."*
*(以旱灾免山西大同等府卫去年秋粮子粒二十三万余石马草四十三万四千余束)*

In contrast, during the Great Drought of the Wanli period, no specific quantities were recorded.
*"The decree ordered that the tax could be exempted at varying levels for the disaster-stricken populations in Shaanxi and Shanxi"*
*(诏狭西山西被灾民屯钱粮蠲折有差)*

As a result, it is difficult to assess the differences in the central government's response intensity between the two drought events. Therefore, we did not rely on the Ming Shilu. *Compendium* reflects local data, which is the specific implementation of the state relief at the county level. Therefore, it also provides valuable insight into state' capacity.

**Responses to technical comments:**
In line 101, "50 million" should be "62 million", see Cao, 2000, pp. 451~452. In addition, Zheng et al. (2014a) speculated that the total population of North China in 1580 was about 52 million (see Table S4 of the ESM), which would be a useful reference.
Response: We appreciate the reviewer's comments. We have revised, citing Cao's work as the reference.

Line 104, "Dan (a weight unit in ancient China)" is not accurate. Dan (石) is an unit of volume in ancient Chinese, and in the Ming Dynasty, the weight of 1 Dan grain is about 60~70 kg.
Response: Thanks. We have made the revision and added the note.

The translation of "大旱/大饥" as "major drought/major famine" is inaccurate (e.g. Tab.1), great or severe might be more accurate.
In line 212, there should be a note on "Guanyin tu (观音土)", which is a kind of clay with fine texture that can be fatal if consumed in excess.
Tab. 3, in "countless starvation deaths (死者甚众), corpses as pillows (死者甚众)", Chinese and English do not correspond, so the original text needs to be checked.
Response: Thanks. We agree. We have made the revision accordingly.

In line 318-321, about the planting scope of American crops in China in the Ming Dynasty, current researchers believe that corn was mainly distributed in southern China at the end of the 16th century (late Ming Dynasty), and the planting area in the north was very small; The potato was probably not yet introduced to China (see Han Maoli, Historical Agricultural Geography of China, vol. 2, 韩茂莉：《中国历史农业地理》中册). The contribution of American crops to the improvement of social response capability in the Wanli reign is not significant.
Response: We appreciate the reviewer's suggestion. More researches (He, 1979; You, 1989; Wang, 2004) have led us to recognize that we had overstated American crops' influence on the northern

regions during the Wanli Drought. As a result, we have removed the section discussing American crops to maintain the accuracy and integrity of our analysis.

**Response to reviewer #3,**

**Responses to specific comments:**

This study presents the impacts and responses of extreme drought events during the ancient China. Broadly speaking, it can help understand the occurrence of famine and provide insights for countries or regions at risk of food shortage. Currently, China is already making a big contribution to world hunger and world food security. What we're more interested in is how historical Chinese famine responses influence modern-day disaster management? which measures are still in use today? I suggest the article should add a brief section about the inheritance of historical response measures.

Response:We appreciate the reviewer's comments. We have added a section in **Discussion 4.2: Inheritance and development of historical disaster experience**, where we explore the impact of historical disaster management experiences in China on modern disaster management from the perspectives of inheritance and development.

**Responses to technical comments:**

Line 318-322: Potatoes were not widely spread and planted until the Qing Dynasty. Does the article exaggerate the role of potatoes?

Response: We appreciate the reviewer's comments. We had overestimated the impact of American crops and removed this section.

Are the AD1s showed in Table 4 and Table 6 same?

Response: Thanks. We have added the formula derivation process and revised the notation to avoid confusion.

References in text should be standardized, for example: in line 355 (Su 2010); in line 394 (Liang, 2008)

Response: Thanks. We have made the revision accordingly.

---

## Author Response (AR2)

**Dear Editors,**

Thank you for your decision and suggestions on our manuscript. We sincerely appreciate the time and effort you have dedicated to evaluating our manuscript. In response to your feedback, we have made the following corrections:

1. We have added data sources for the digital elevation in Fig.1. At the same time, we choose the option (b), that you can add a statement that some figures contain disputed territories.
2. We have added explanations for Fig.3 and Fig.6.
3. We have labelled Fig.7 and updated in the preceding text accordingly.
4. We clarified that "$D$" and "$F$" refer to the drought grade and famine grade of drought-affected counties in one drought events. These are different from $AD_j$ and $AF_j$. Necessary corrections have been made.

We have uploaded the revised manuscript file. Thank you again for your valuable suggestions.

Thanks again!

Sincerely,

Authors.